# Ion complexation waves emerge at the curved interfaces of layered minerals

Michael L. Whittaker [1,2 ✉], David Ren [3], Colin Ophus [4], Yugang Zhang [5], Laura Waller [3], Benjamin Gilbert [1,2] & Jillian F. Banfield [1,2]

Visualizing hydrated interfaces is of widespread interest across the physical sciences and is a particularly acute need for layered minerals, whose properties are governed by the structure of the electric double layer (EDL) where mineral and solution meet. Here, we show that cryo electron microscopy and tomography enable direct imaging of the EDL at montmorillonite interfaces in monovalent electrolytes with ångstrom resolution over micron length scales. A learning-based multiple-scattering reconstruction method for cryo electron tomography reveals ions bound asymmetrically on opposite sides of curved, exfoliated layers. We observe conserved ion-density asymmetry across stacks of interacting layers in cryo electron microscopy that is associated with configurations of inner- and outer-sphere ion-water-mineral complexes that we term complexation waves. Coherent X-ray scattering confirms that complexation waves propagate at room-temperature via a competition between ion dehydration and charge interactions that are coupled across opposing sides of a layer, driving dynamic transitions between stacked and aggregated states via layer exfoliation.

[1] Energy Geosciences Division, Lawrence Berkeley National Laboratory, 94720 Berkeley, CA, USA. [2] Department of Earth and Planetary Science, University of California, 94720 Berkeley, CA, USA. [3] Department of Electrical Engineering and Computer Sciences, University of California, Berkeley, CA 94720, USA. [4] National Center for Electron Microscopy, Molecular Foundry, Lawrence Berkeley National Laboratory, 94720 Berkeley, CA, USA. [5] Center for Functional Nanomaterials, Brookhaven National Laboratory, Upton, NY 11973, USA. ✉email: mwhittaker@lbl.gov

Layered minerals control carbon, water, nutrient, and critical element transport in the lithosphere[1–6], lubricate fault slip[7–9], and promote cloud formation[10] through interactions among charged, hydrated interfaces[11]. Consequently, intermolecular forces that control ion and water distributions at the mineral-aqueous interface have been widely investigated[12–16] via measurements of the potential drop at the interface[17], the change in orientation and density of water molecules[18–20], and ordering of water and counterions[21–25]. Electric double layer (EDL) models that form the basis of Derjaguin Landau Verwey Overbeek (DLVO) theory accurately describe mica surfaces in monovalent (e.g., $Li^+$, $Na^+$, $K^+$) electrolyte[26] but provide an incomplete description[27] of clay mineral behavior such as swelling pressure[28]. Discrepancies between the measured and simulated properties of bulk and clay-sized interfaces have been attributed to differences in molecular structure[29] or non-DLVO forces[30], but geometrical and dynamical factors may also play a significant role.

Theories of swelling in layered mineral systems have relied on the assumption that layers have geometries consistent with the Derjaguin approximation, which reduces the effective force between two bodies to an interaction between planes. This is a seemingly logical choice for layered minerals, which has been shown to apply in limiting cases[28]. However, images of fully hydrated, polydisperse layered mineral suspensions show that stacks of layers can swell[31,32], exfoliate[33], bend[34], aggregate[35], and adopt microstructures for which the Derjaguin approximation may not hold. These structures are often characterized as glasses or gels[36,37] and do not exhibit equilibrium macroscopic behavior[38–41], suggesting that the physical consequences of violating the Derjaguin approximation may result in quantifiable differences in the distributions of water and electrolyte at interfaces that change over time.

Images of electrolyte distributions where liquids and solids meet, the strongest constraint on theories of hydrated interfaces, have remained elusive due to the challenge of simultaneously imaging both solid and liquid with high resolution. Thus, there has been no direct experimental link between the microstructure of mineral layers and the ion distributions around them. Here, we show using cryo electron microscopy (cryoEM) that the interfacial distribution of electrolyte is strongly influenced by the topology of exfoliated layers and is inextricably coupled to interactions among neighboring layers. Using a learning-based reconstruction algorithm[42] for cryo electron tomography (cryoET) we determine the average ion density at montmorillonite (Mt) interfaces and show that it is strongly dependent on curvature. Stacks of curved layers, or tactoids, were observed to aggregate and exfoliate with avalanche dynamics in X-ray photon correlation spectroscopy (XPCS), which is a direct result of the coupling between layer topology and ion (re)distribution among interacting EDLs in the general case that a mineral interface is not at equilibrium.

## Results and discussion

We find that layer dynamics in montmorillonite (Mt) suspensions at low-electrolyte conditions are largely consistent with DLVO theory, and provide a reference point for understanding the qualitatively new behavior that emerges at elevated electrolyte concentrations. Structural dynamics of Li-Mt, Na-Mt, and K-Mt suspensions were investigated in situ using the two-time correlation function, $\chi(q, t_1, t_2)$, of coherent scattering intensity from XPCS (Fig. 1). All systems exhibited scattering intensity correlations ($\chi > 1$) between two different measurement times, $t_1$ and $t_2$, at scattering vectors, $q$, corresponding to the approximately 50–250 nm distribution of Mt layer sizes. The correlated motion

of layers that are separated by 1–200 nm of solution (Figs. S1–2) is a clear indication that the electrolyte couples their movement over distances comparable to the dimensions of the layers themselves. However, layers interacted and moved differently depending on the concentration and identity of the electrolyte at otherwise identical mineral volume fractions (in this case, approximately 2%), revealing new interfacial phenomena that ultimately control swelling mineral behavior.

Na-Mt suspended in ultrapure water exhibited structural correlations that increased steadily over time ($\chi \rightarrow 1.2$), shown in Fig. 1a for the case at $q = 0.0025$ Å$^{-1}$ (250 nm, full range of $q$ in Fig. S4). Static in situ X-ray scattering (Fig. 1b) confirmed that the average microstructure of this suspension is nematic, with layers that are planar, parallel and separated by an average interlayer distance of $d = 58$ nm along the nematic director field (arrow, Fig. 1b). The Debye screening length, $\kappa^{-1}$, the characteristic distance over which the surface potential, $\psi$, decreases, is also approximately 58 nm under these conditions. $\kappa^{-1}$ is fixed by the background electrolyte concentration, which in our case is set by the solubility limit[33] of amorphous silica (Supporting Information). Thus, the nematic structure of Na-Mt is maintained by repulsion of overlapping EDLs, a commonly observed phenomenon in clay suspensions[43] that is quantitatively predicted by DLVO theory (Figs. S1–2) and supports the validity of the Derjaguin approximation in this regime.

We find that nematic layers have effectively no mobility at distances below 58 nm, as determined from relaxation times extracted from the one-time autocorrelation function, $g_2(q, t)$, derived from $\chi(q, t_1, t_2)$ (Fig. 1g, h, Supplementary Information). Thus, the gradually slowing dynamics in Fig. 1a correspond to the gradual alignment of the nematic director field (Fig. 1c) for layers with translational diffusivities, $D$, of $1.7 \times 10^3$ Å$^{-2}$ s$^{-1}$ (Fig. 1h, Figs. S3–4) above 58 nm separation and essentially zero at smaller interlayer distances, where EDL repulsion restricts layer motion. We call this state the nematic glass because its structure is ordered along a single axis and the interactions between layers are dominantly repulsive. Because the EDL repulsion energy is relatively small under these conditions (Supplementary Information), the act of moving the sample capillary into the beam path is sufficient to disrupt the nematic structure and trigger the slow relaxation towards nematic order (Fig. 1c)

Static in situ X-ray scattering confirms that tactoids, stacks of multiple layers with an average interlayer spacing of $d = 1.9$ nm, are a prevailing structural unit for Na-Mt in 1 M NaCl (Fig. 1e). This is consistent with the increasing role of attractive van der Waals interactions[32] at elevated electrolyte concentrations, because increasing electrolyte concentration decreases $\kappa^{-1}$, and therefore $\psi$, reducing the strength of repulsive osmotic interactions. Particle dynamics for these tactoids are markedly different from the nematic glass. Alternating correlated and uncorrelated states, which manifest as local extrema in $\chi(t_1)$ for constant $q$ and $t_2$, are observed under these conditions (Fig. 1d). Average tactoid diffusivities are 2.5 times higher than single nematic layers ($4.3 \times 10^3$ Å$^{-2}$ s$^{-1}$, Figs. S4–7), despite comprising an average of ~5 layers[33] and therefore having fivefold greater mass. Furthermore, tactoid motion is uninhibited at small distances (Fig. S7), consistent with minimal EDL repulsion between tactoids. This dynamical behavior is consistent with particles that are much more mobile than those in nematic glasses. However, it is very surprising, given that these conditions are well within the coagulated or flocculated region of reported Mt phase diagrams[43] where layers are expected to be kinetically arrested to a much greater extent than the nematic glass. We find that the degree to which Mt dynamics appear arrested depend strongly on the scale at which they are investigated, and are fundamentally controlled by cation binding at the mineral interface.

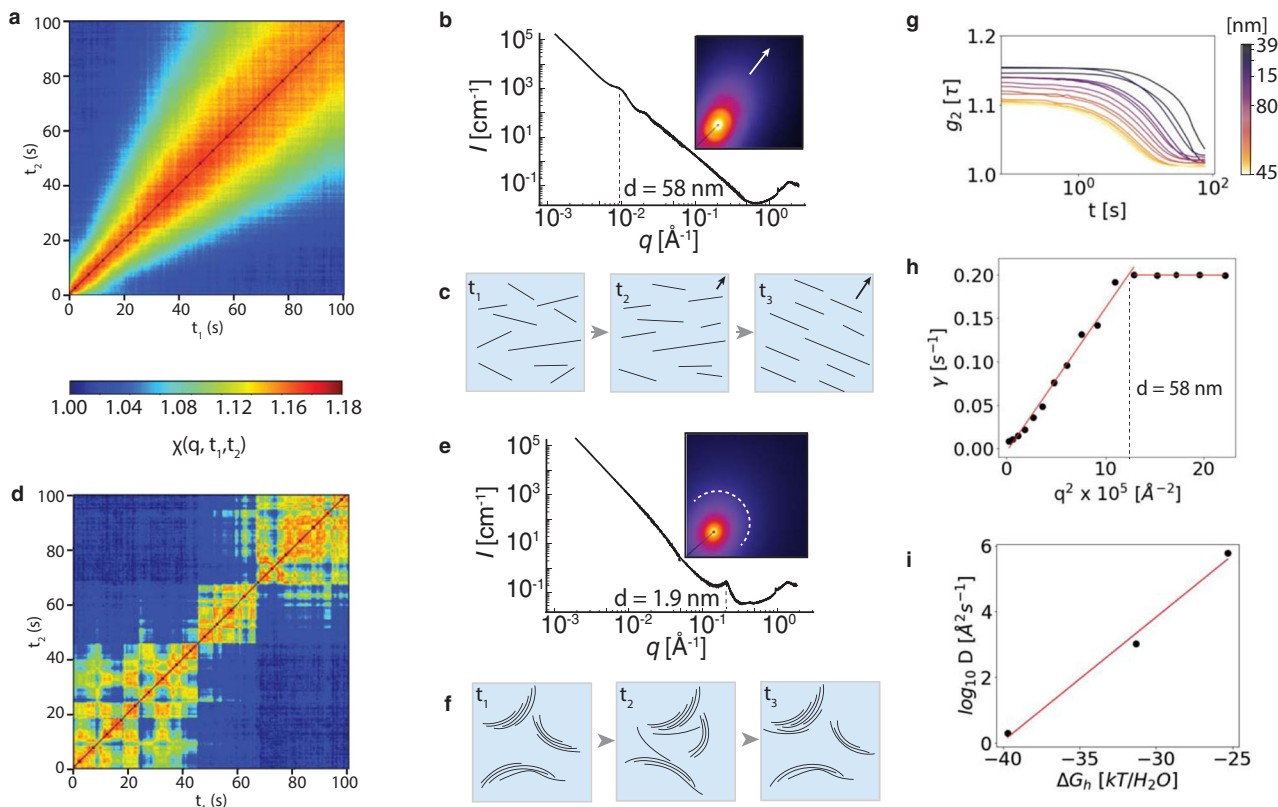

**Fig. 1 Structural evolution of hydrated Mt. a** XPCS two-time correlation of Na-Mt in ultrapure water. **b** Static X-ray scattering and **c** schematic of nematic director alignment. **d** Two-time correlation of Na-Mt in 1 M sodium chloride. **e** Static X-ray scattering and **f** schematic of tactoid avalanche transitions. **g** $g_2$ from two-time data in (**a**) and (**h**) fit of the relaxation time using Eq. S5. **i** Linear relationship between the hydration energy per water molecule for Li, Na, and K[44] and $\log_{10}$ of the diffusivity determined from relaxation plots such as (**h**) in 1 M electrolyte.

We find that suspension dynamics varied dramatically depending on the cation identity, which departs from expectations based on measurements of monovalent ion distributions at mica surfaces[26]. Tactoid diffusivities (Figs. S4–9) increase exponentially with the cation hydration energy[44] ($\Delta G_h$) in the order K > Na > Li (Fig. 1i). This strongly suggests that ion (de)hydration is a dominant control on interfacial interactions. Remarkably, the behavior of the mineral layers at length scales over four orders of magnitude larger than the scale of the hydration interaction itself are impacted by ion (de)hydration. Thus, the hydration energy of the cation must have a strong influence on the mineral surface charge, and therefore ψ (Supplementary Information) which influences the mobility of the layers. When the hydration energy is comparable to the drop in the Stern layer potential across the plane of ions bound at the mineral surface (Supplementary Information), fluctuations in solution chemistry or the topology of the surface (e.g., curvature) can lead to dynamic (de)hydration. The result is that tactoids dynamically oscillate between 'interacting' and 'diffusing' states that become more or less correlated over the 10 × 10 μm size of the beam at a rate that depends on the cation hydration energy. In other words, tactoid aggregates dynamically assemble and dissolve.

To understand these remarkable differences in behavior that appear to be caused by the hydration of cations at interfaces, we obtained direct evidence of ion distributions at the interfaces of Na-Mt layers and tactoids using low-dose cryo electron microscopy (cryoEM) (Fig. 2). We demonstrate that it is possible for the first time to directly visualize ions in inner and outer-sphere binding locations at hydrated mineral surfaces by imaging near zero defocus, thereby maximizing the resolution, and using an energy filter to enhance absorption contrast from crystalline

layers oriented along the beam axis (Fig. 2). Anions are excluded from the interlayer space of crystalline hydrate tactoids in which layers are separated by less than 2 nm[45], simplifying the interpretation of cryoEM image contrast by allowing only contributions from mineral and sodium ions. These data reveal differences in binding on external and internal tactoid surfaces and suggest a link between curvature, hydration and ion complexation that gives rise to asymmetric layer charge on convex and concave sides.

We focus on two of the four layers within a representative tactoid, labeled[1] (Fig. 2c), and[2] (Fig. 2d). This tactoid is curved in- and out-of the plane of the image, with the concave side oriented downward and the convex side at the top. Lattice peaks in the Fourier transformation of regions below, but not above, the tactoid confirm its curved geometry (Fig. S11). On the exterior, convex side of the tactoid (Fig. 2a, top of layer 1), we observe localized contrast at 6.4 Å from the layer midplane, consistent with sodium ions bound as outer-sphere complexes with water (Fig. 2e–g). The observation of two high-contrast planes at 11.1 Å and 14.4 Å agree with atomistic simulations predicting structured diffuse layers containing a mixture of sodium and chloride ions at the Mt interface[46,47]. On the interior, concave side of layer 1, interlayer sodium ions formed primarily inner-sphere complexes at fixed positions 5.0 Å from the layer midplane (Fig. 2e–g).

Surprisingly, asymmetric counterion binding persists throughout the tactoid. The convex side of layer 2 shows predominantly inner-sphere binding while the concave side shows mixed inner- and outer-sphere binding (Fig. 2h–j). This polarization persists across all layers in the stack (Fig. 2b). The observation of asymmetric counterion complexation on internal and external sides of the Mt tactoid has not been explored by

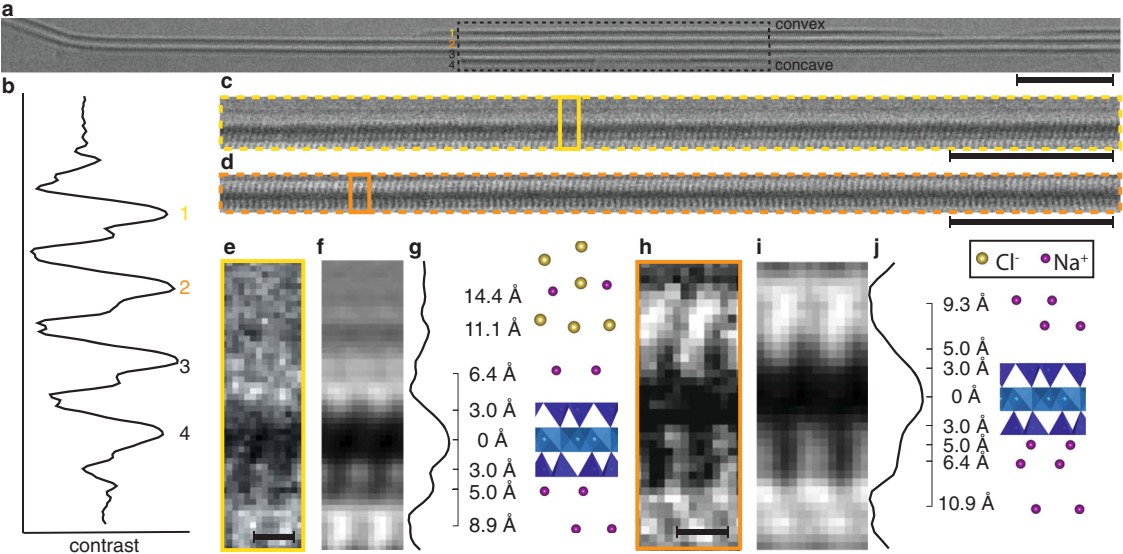

**Fig. 2 CryoEM structure of Na-Mt and aqueous interfacial solution. a** Na-Mt tactoid with four layers, each separated by 19 Å (scale bar represents 20 nm). **b** Contrast profile integrated along box in (**a**). **c** Layer 1 enlarged. **d** Layer 2 enlarged. **e** Two unit cell wide region from layer 1 showing the interface with the aqueous solution (scale bar represents 5 Å). **f** Average unit cell, duplicated for comparison to (**e**). **g** Contrast profile normal to layer 1, showing distances to local contrast minima relative to the layer midplane and proposed structural model of Mt and electrolyte structure adjacent. **h** Two unit cell wide region from layer 2. **i** Average unit cell, duplicated for clarity. **j** Contrast profile normal to layer 2, with structural model of Mt and counterion structure adjacent.

prior experiment or simulation to our knowledge. We term this phenomenon a 'complexation wave' because inner- and outer-sphere hydrated ion complexes propagate throughout a tactoid with the same periodicity.

A Grahame triple layer adsorption model (Supplementary Information) captures the effect of curvature, $H$, on ion binding when explicitly accounting for the bending free energy[48], $\Delta G_b(H)$, and ion hydration free energy in addition to electrostatic forces. Bending imposes a locally varying electrostatic potential that changes the strength of electrostatic interactions relative to the planar electrostatic potential ($\psi$), effectively acting as a mineral activity coefficient. The mineral activity varies with bending because $H$, which is a signed quantity and therefore may differ on opposing sides of curved layer[48], causes a greater fraction of ions, $\alpha$, to be bound on the concave side versus the convex side

$$\alpha = \frac{1}{1 + \frac{C}{a}e^{\beta\left(e\psi - \Delta G_h - \Delta G_b(H)\right)}} \qquad (1)$$

where $a$ is the water activity, $e$ is the elemental charge, and $\beta = -1/kT$. Because $\psi$ is fundamentally an osmotic potential, and both mineral and electrolyte are osmolytes, this expression for $\alpha$ quantifies the inherent osmotic coupling between bulk solution ($C$, $a$, $\Delta G_h$) and mineral ($\psi$, $\Delta G_b(H)$). Equation 1 is transcendental because $\psi$ and $G_b(H)$ also depend on $\alpha$, and this has important physical ramifications: complexation waves form because ions can reside in multiple binding configurations with equivalent energies, depending on the local degree of curvature. Likewise, changes in $H$ can also be induced by local changes in electrolyte concentration that change the proportions of inner- and outer-sphere complexation. This nonlinear coupling between solution and surface is precisely the reason that the Derjaguin approximation breaks down for swelling clays. In analogy to the Poisson-Boltzmann equation used to find $\psi$ for planar layers, Eq. 1 can be solved graphically or numerically (Figs. S14–17).

Despite the ability to directly interpret thin electron-optical slices of the sample by energy-filtered imaging near zero defocus, there remains ambiguity about assigning a three-dimensional

(3D) complexation wave structure to a two-dimensional (2D) projection image. In order to confirm the observation of asymmetric ion complexation, and describe its effects on the suspension microstructure at larger length scales, we applied cryo electron tomography (cryoET) to Li-Mt suspensions and validated these observations with cryoET simulations (Figs. S11–13).

Mineral and electrolyte distributions were resolved in unprecedented 3D detail by implementing a learning-based reconstruction algorithm[42] that accounts for multiple electron scattering from low-dose images acquired at multiple defocus values at each tilt angle. This enabled the recovery of both the amplitude and phase of the electron exit wave in three dimensions, revealing interfacial structures across thousands of mineral layers with the highest isotropic real-space resolution from single cryo-frozen samples reported to date, to our knowledge, 3.64 Å, over a 1.02 μm × 0.79 μm × 0.36 μm field of view (Fig. 3).

At a mineral volume fraction of 2% and electrolyte concentrations of $C = 0.1$ M and 0.75 M the Li-Mt suspensions are composed of mixtures of exfoliated layers and stacked tactoids separated by electrolyte solution (Fig. 3a, d). All Mt layers are curved, despite having chemical compositions that are symmetrical about the layer midplane and that exhibit no spontaneous curvature (e.g., nematic microstructures in Fig. 1b). We find that the degree of curvature differs between the two electrolyte concentrations (Fig. 3b, e), which is consistent with differing counterion complexation profiles (i.e, charge distribution) on opposing sides expected from Eq. 1.

Radial integration of the cryoET Fourier-transformation amplitude is directly comparable to the X-ray structure factor (Fig. 3c, f). However, the difference is that both the phase and amplitude of the cryoET data are known by virtue of recording the data in real space first, allowing us to unambiguously interpret the structure factor peaks. For example, a peak in the cryoET structure factor between scattering vectors $q = 0.50$–$0.57$ Å$^{-1}$ (Fig. 1c, f) corresponds to an average layer thickness, $\langle t \rangle$, that includes the aluminosilicate layer and hydrated lithium counterions at the interface, but not bulk electrolyte. The thickness of an exfoliated layer is 12.6 $+1.7/-1.8$ Å in 0.1 M lithium chloride

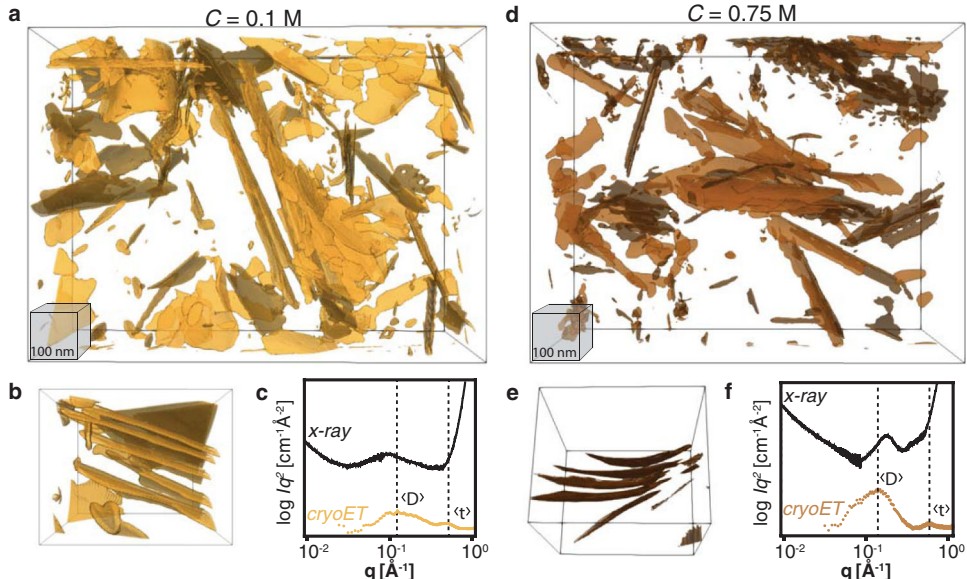

**Fig. 3 Li-Mt suspensions viewed with cryoET.** Isosurfaces of Li-Mt layers in 0.1 M (**a**) and 0.75 M (**d**) lithium chloride. **b**, **e** Subvolumes from (**a**) and (**d**), showing osmotic hydrate stacks. Comparison of structure factors from X-ray scattering and cryoET in 0.1 M (**c**) and 0.75 M (**f**), showing larger average interlayer spacing and larger layer thickness at lower electrolyte concentration.

and 11.0 Å +1.0/−1.9 in 0.75 M lithium chloride. Thicker layers at low-electrolyte concentration are thus indicative of a greater fraction of fully hydrated lithium ions that reside further from the layer midplane, while conversely, thinner layers at elevated electrolyte concentration are direct evidence of a higher fraction of lithium ions that make inner-sphere complexes with the mineral interface. This is in accordance with the expectation of Eq. 1, which predicts that higher electrolyte concentration drives complexation equilibria towards partially dehydrated inner-sphere complexes.

The real-space structure of the EDL and its dependence on curvature were first quantified by extracting and averaging the reconstructed absorbance magnitudes normal to the surface of Mt layers. Representative exfoliated Mt layers with low and intermediate curvature (Fig. 4a, b) are compared with Mt layers exhibiting higher curvature in an osmotic hydrate (Fig. 4c). Statistical analysis of between $5.3 \times 10^4$ and $1.3 \times 10^5$ absorbance profiles taken normal to the layers at each midplane voxel revealed features below the nominal voxel resolution (Fig. 4d–f), in analogy with the common practice of particle- or subtomogram averaging in cryoEM of biological macromolecules[49]. The resulting averaged ion-density profiles reflect some expected aspects of EDL models of layer silicates. For example, a region of low absorbance extends approximately 5 nanometers from the layer midplane (Fig. 4d–f), arising from the structured water and cation-rich interfacial structure[47] and the depletion of chloride ions that are repelled from the negatively charged mineral. Thus, absorbance profiles starting from the layer midplane and moving into the bulk solution can be attributed to dominant contributions from mineral, lithium, and chlorine, respectively.

In contrast to existing models, however, real-space ion-density profiles confirm a prominent role for layer curvature in modulating interfacial ion distributions. Compared to the low-curvature layer (Fig. 4d), both high-curvature layers show asymmetry in the anion depletion region directly adjacent to the mineral surface and in the anion distributions at distances up to 15 nm from the mineral (Fig. 4e, f). Thus, concavity sequesters counterions over appreciable distances near exfoliated layers. This shift in anion density distribution is correlated with an increase in

the lithium content at the surface, which we show is attributable to increased cation complexation and concomitant reduction in surface charge.

In order to distinguish overlapping contributions to the measured absorbance profiles, we also applied non-negative matrix factorization (NNMF) to all profiles for a given layer. Lithium-ion complexation at the mineral interface was quantified from the second of two NNMF factors (Fig. 4g–i). The first factor accounts for the absorbance of the layer itself and the inner-sphere complexation of hydrated lithium, with a full width at half maximum (FWHM) at 4.2 Å from the midplane[13]. The second factor is consistent with outer-sphere complexation of partially dehydrated layer at 5.8 Å from the midplane. This result supports the reciprocal-space interpretation of the cryoET structure factor from Fig. 3c, f, in which two distinct binding configurations with variable concentrations contribute to the changing thickness of a layer with electrolyte concentration.

We observe that outer-sphere lithium complexation in $f_2$ is asymmetrically distributed, with greater concentrations on the convex side relative to the concave side. Due to the strong absorbance from the layers, contributions from inner-sphere Li complexes could not be directly quantified. However, asymmetric outer-sphere complexation increases with increasing curvature (Fig. 2g–i), a clear demonstration that both inner- and outer-sphere complexation states coexist and that their relative proportions are dependent on layer curvature. Differential charge on opposing sides of a layer due to complexation waves influences interlayer forces but also depends on the bulk electrolyte identity and concentration; complexation waves are sustained on isolated layers when EDLs repel, but couple across tactoids as the complementary convex and concave sides of neighboring layers attract.

Together, these findings demonstrate that complexation waves appear over a wide range of conditions in layered mineral systems that arise from the exchange of elastic, electrostatic and hydration energy as ions partition from the bulk electrolyte, complex with the mineral layer and induce it to bend. Prior observations of delamination and restacking during Na⁺/K⁺ ion exchange[33] demonstrated that dynamic rearrangements of smectite tactoids

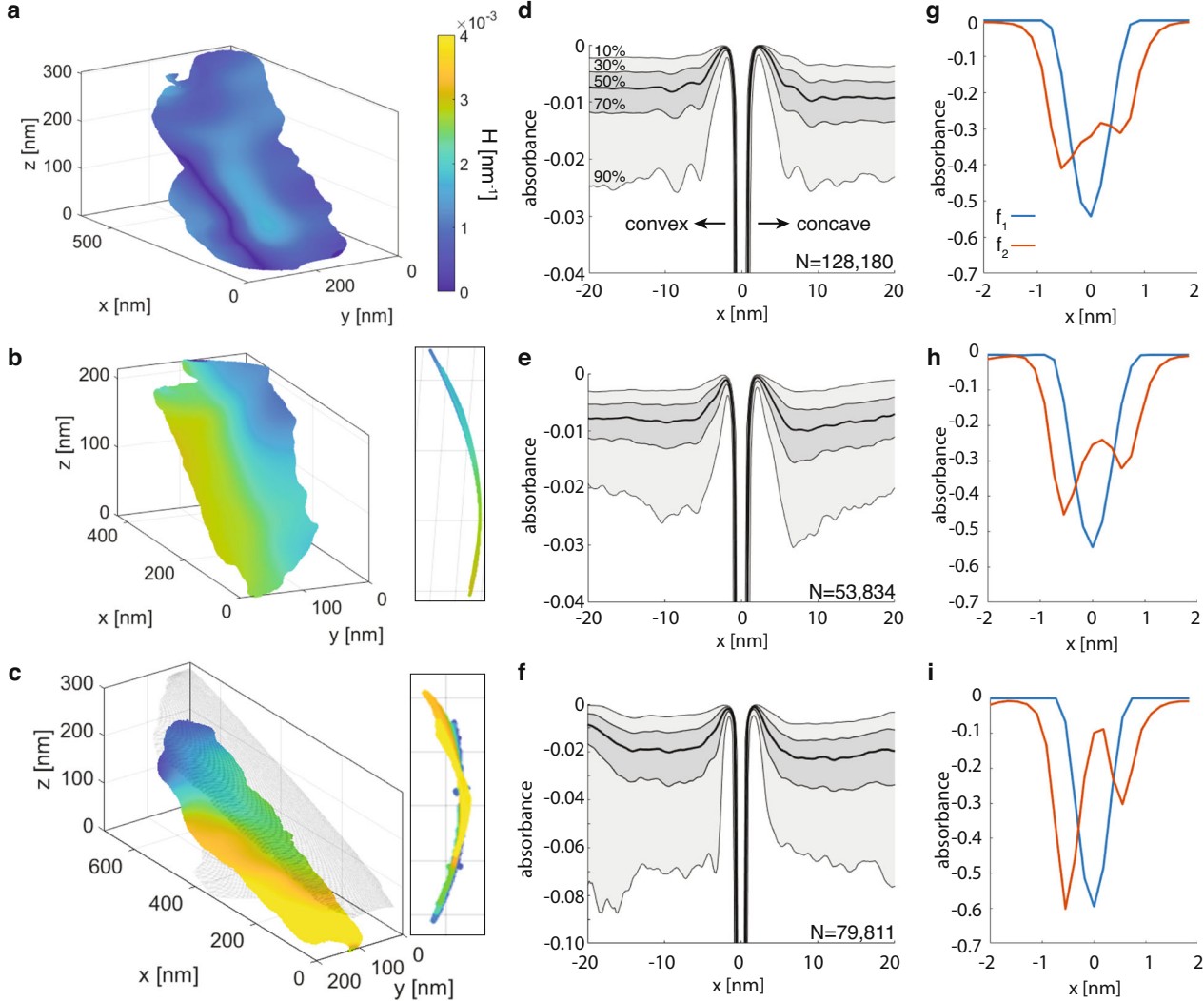

**Fig. 4 Effect of layer curvature on montmorillonite-electrolyte ion-density profiles in 0.1 M lithium chloride. a** Single exfoliated layer with low curvature. **b** Single layer with intermediate curvature. **c** Highly curved layer in an osmotic hydrate stack. Colored according to local curvature, H (Supplementary Information). **d** Average ion-density profiles from within 20 nm of convex and concave sides of the low-curvature layer. Shading indicates the percentile of observations accounted for from 128,180 individual profiles (i.e., '10%' includes the 10% lowest absorbance points at each distance). **e** Ion-density profile from the intermediate-curvature layer. **f** Average ion-density profiles between stacked layers (note difference in scale). **g** Non-negative matrix factorization (NNMF) of all absorbance profiles in (**d**), showing first two factors, $f_1$ (Mt) and $f_2$ (Li). **h** Increasingly asymmetric $f_2$ in NNMF profile of curved layer in (**b**). **i** Highly asymmetric NNMF $f_2$ for curved stacked layer.

are coupled with ion intercalation. Here we show that tactoid aggregation, delamination, and restacking also occur in homo-ionic solutions and arise from the propagation of complexation waves. Layer diffusivities are at least 10 orders of magnitude smaller than those of ions or water. Abrupt transitions between aggregated and dissolved structures occur when (relatively slow) layer bending is required to resolve an osmotic gradient that results from the (relatively rapid) accumulation of electrolyte near a transiently curved interface.

Avalanche transitions, such as those between aggregated tactoids, arise when nonlinear dynamics generate disproportionately large responses from small perturbations[50]. The surprising observation that tactoid dynamics at length scales of hundreds of nanometers are controlled by the hydration energy of a cation confirm that surface complexation is the small perturbation that, through the collective interaction of neighboring complexes and their influence on layer curvature, generates a disproportionately large response via the interaction of layers throughout the sus-pension. These transitions are the result of a violation of the

Derjaguin approximation, such that layer geometries and binding configurations are not in global equilibrium, but find a local energy minimum near the thermal energy.

We expect that complexation waves, which fundamentally result from the breaking of interfacial symmetry, are a general phenomenon at curved interfaces that couple interfacial strain and charge distributions. Unprecedented insights into the behavior of layered minerals confirm that cryoET and cryoEM open a new window into the structure of hydrated interfaces that is essential for the quantitative interpretation of aqueous interfacial phenomena that underlie myriad geochemical processes.

# Methods

**Materials**. Wyoming montmorillonite (SWy-3), obtained from the Source Clays Repository of The Clay Minerals Society (http://www.clays.org/sourceclays_data.html), was used throughout this study. Aqueous solutions of lithium chloride, sodium chloride, potassium chloride were prepared from reagent-grade salts and used for homoionization of the clay using standard techniques[1].

**CryoET/cryoEM.** Suspensions of Li-Mt and Na-Mt with mineral concentrations of 5 mg/mL were deposited as 3 μL aliquots onto 200-mesh lacy carbon Cu grids (Electron Microscopy Sciences) which had been glow-discharged in air plasma for 15 seconds. Excess solution was removed by automatic blotting (1 blot for 10 s, blot force 10 at 95% relative humidity) before plunge-freezing in liquid ethane using an automated vitrification system (FEI Vitrobot). Imaging was performed with a Titan Krios TEM operated at 300 kV, equipped with a BIO Quantum energy filter. Images were recorded on a Gatan K3 direct electron detecting camera with a pixel size of 0.91 Å/pixel in superresolution mode for cryoET and 0.75 Å/pixel for cryoEM. Imaging was performed under cryogenic conditions using a low electron dose of 121 e$^-$/Å$^2$ for cryoEM images and 1100 e$^-$/Å$^2$ for cryoET. Dose-fractionated movies with a total dose of 3 e$^-$/Å$^2$ were acquired at tilt angles ranging from ±60° in 1° increments and defocus values of −75, −200, and −550 nm at each tilt angle in a dose-symmetric scheme starting at 0° and 0 nm defocus using a custom script in SerialEM software.

**CryoET reconstruction.** Dose-fractionated movies were gain corrected and aligned in RELION3.0 and summed to form intensity images. Intensity images were normalized, Fourier downsampled to 3.64 Å/pixel, pre-aligned using an in-house code written in MATLAB, and the alignment was refined using IMOD. Tomographic reconstruction was performed iteratively using the method developed by Ren et al[2]. to model HRTEM contrast from multiply scattering samples. We used a custom implemented Python library, which supports GPU computation (https://github.com/yhren1993/PhaseContrastTomographySolver) to perform reconstructions on the Lawrence Berkeley National Lab High Performance Computing Clusters in a distributed fashion. The final reconstruction had an isotropic voxel size of 3.64 Å.

**CryoET segmentation.** The clay sheets in the reconstructed cryoET absorption volumes were segmented using custom codes written in Matlab. The goal was to reduce the voxels corresponding to clay layers to a set of "sheets" which were defined as a cloud of points representing a 2D sheet embedded in 3D space, each with an associated vector representing the sheet surface normal. We used Fast Fourier Transforms to efficiently compute the correlation between an isotropic orientation kernel and each voxel in the reconstruction volume, then applied a two-level threshold[1] a global threshold by using a minimum value for the correlation signal, and[2] voxels with correlation signals greater than at least 18 neighboring voxels (out of a possible 26 neighbors). Mean surface curvature 2H was estimated by fitting the 2D parabolic local surface to the expression

$$2H = \frac{\left(1 + S_x{}^2\right)S_{yy} - 2S_x S_y S_{xy} + \left(1 + S_y{}^2\right)S_{xx}}{\left(1 + S_x{}^2 + S_y{}^2\right)^{3/2}}$$

where $S_x$ and $S_y$ are the first order spatial derivatives of the surface, and $S_{xx}$, $S_{yy}$, and $S_{xy}$ are the second order derivatives.

**Absorbance profile quantification.** Absorbance profiles were measured by mapping the normal vectors from segmented layers to the unsegmented data and using trilinear interpolation to quantify the absorbance at a given distance along each normal vector. Non-negative matrix factorization (NNMF) implemented in Matlab was used to establish two absorbance profile factors across all profiles in a given layer.

**X-ray scattering.** X-ray scattering was performed at beamline 5ID-D of the Advanced Photon Source at Argonne National Laboratory in order to obtain high photon fluxes necessary for time-resolved experiments. Small-, medium-, and wide-angle X-ray scattering (SAXS/MAXS/WAXS) was collected simultaneously on three Rayonix charge-coupled device (CCD) detectors with sample−detector distances of 8505.0, 1012.1, and 199.5 mm, respectively. The wavelength of radiation was set to 1.2398 Å (10 keV), resulting in a continuous range of scattering vector, $q = 0.017$–$4.2$ Å$^{-1}$.

**XPCS.** XPCS experiments were conducted at the Coherent Hard X-ray (CHX) beamline 11-ID at the National Synchrotron Light Source II (NSLS-II), Brookhaven National Laboratory. The X-ray energy was 9.65 keV ($\lambda = 1.285$ Å) with energy resolution $\Delta E/E \approx 10^{-4}$ from a Si111 double crystal monochromator. A partially coherent X-ray beam with a flux at the sample of ~$10^{11}$ photons/s and a focused beam size of $10 \times 10$ μm$^2$ was achieved by focusing with a set of Be Compound Refractive Lenses and a set of Si kinoform lenses in front of the sample. The sample was loaded in a wax-sealed glass capillary mounted on the sample stage. The coherent scattering pattern was recorded in transmission small angle scattering geometry by using a photon-counting pixelated area detector (Eiger X 4M Dectris Inc.) located 16.03 meters away from the sample with a 75 μm × 75 μm pixel size. The X-ray radiation dose on the sample was controlled by a millisecond shutter and filters of different thickness of silicon wafers. The data acquisition strategy was optimized to ensure that the measured dynamics and structure are dose independent. The XPCS data analysis were conducted by using software developed at CHX, NSLS-II. A q range of $Q = 0.0015$–$0.9$ Å$^{-1}$, corresponding to length scales

of 0.7−420 nm, captures the lateral dimensions and interlayer spacings of all layers. The two-time correlation function, $\chi(Q, t_1, t_2)$, equals unity if there is no correlation between X-ray scattering intensities at a scattering vector $Q$ for an initial time $t_1$ and second time $t_2$, and approaches 1.2 for intensities that are unchanged between the two-time points.

## Data availability

The datasets generated during and/or analyzed during the current study are available in the Zenodo repository, https://doi.org/10.5281/zenodo.4574610.

## Code availability

Our code cryoET reconstruction code is available at (https://github.com/yhren1993/PhaseContrastTomographySolver).

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

## Acknowledgements

We thank Steven Weigand for technical assistance with X-ray scattering measurements and Dan Toso for technical assistance with the FEI Titan Krios. We thank Laura Lammers, Christophe Tournassat and Chenhui Zhu for helpful discussions. U.S. Department of Energy, Office of Science, Office of Basic Energy Sciences, Chemical Sciences, Geosciences, and Biosciences Division, through its Geoscience program at LBNL under Contract DE-AC02-05CH11231. Work at the Molecular Foundry was supported by the Office of Science, Office of Basic Energy Sciences, US. Department of Energy Contract DE-AC02-05CH11231. C.O. acknowledges support from the U.S. Department of Energy Early Career Research Program. Portions of this work were performed at the Advanced Photon Source, a US DOE Office of Science User Facility operated for the DOE Office of Science by Argonne National Laboratory under Contract DE-AC02-06CH11357, at DND-CAT supported by Northwestern University, E.I. DuPont de Nemours and Co., and The Dow Chemical Company, and data were collected using an instrument funded by the National Science Foundation under Award 0960140.

## Author contributions

Conceptualization: M.L.W. conceived the study design, performed cryoEM, cryoET and X-ray scattering experiments, analyzed data, and developed alignment and segmentation algorithms. D.R. and L.W. developed alignment and reconstruction algorithms and DR analyzed the cryoET data. CO developed alignment, segmentation, and reconstruction algorithms and analyzed cryoEM and cryoET data. Y.Z. performed XPCS experiments and analyzed the data. B.G. and J.F.B. conceived the study design. All authors contributed to writing the manuscript.

## Competing interests

The authors declare no competing interests.
