## [Peer Review File · Nature Communications]

Editorial Note: This manuscript has been previously reviewed at another journal that is not operating a transparent peer review scheme. This document only contains reviewer comments and rebuttal letters for versions considered at Nature Communications

REVIEWERS' COMMENTS

Reviewer #1 (Remarks to the Author):

I appreciate the authors efforts to improve the manuscript. But I still do not understand the structure seen in the XPCS time-time correlation plots. For dynamical samples at steady state conditions (e.g., diffusion of spheres in a liquid) the time-time correlation patterns do not change with time. Relaxation dynamics of a glass should lead to slow variations on the time-time correlation functions. The only cases for which I have seen such dramatic behavior in time-time correlation functions is for samples that are driven out of equilibrium by an event (i.e., quenching the sample temperature to drive a phase transition) and then the evolving time-time correlation data after the event reflects the subsequent evolution of the sample. In this experiment, the only event imposed on the sample (to my understanding) is the illumination of the X-ray beam.

The data shown in the rebuttal appear to be consistent with this picture. The plots of total transmitted intensity vs. time in Figures 3-5 show that this signal systematically increases with time. The data also show that the fractional change in intensity becomes larger both for increasing beam flux (from 4% to 100%) and for increasing exposure times (e.g., 1 ms to 100 ms). This suggests that there are significant changes in the sample structure that are driven by the X-ray beam. I believe that this is the source of the complex changes seen in the time-time correlation functions, as opposed to any intrinsic dynamics associated with the electrostatic adsorption of ions.

I do not want to be unfair to the authors. There is a disagreement with respect to the significance of the results. I suggest that another reviewer look at the results to provide another opinion.

Reviewer #2 (Remarks to the Author):

This is not an easy paper to read. The data clearly show that there is an electrolyte dependent change in the samples. This is shown clearly by the qualitative difference in the XPCS data and the detailed analysis of the cryoEM results. It is also clear that there is a non-equilibrium aging behavior of the low concentration data as seen by the increasing correlation times. One question is "what starts this behavior?". Is it something in the preparation of the samples or is it their introduction into the x-ray beam. This increasing correlation time with aging time does not seem so evident in the higher concentration data, at least when I try to picture "averaging" over the fast structures to see the long time behavior. Clearly, something unusual is happening in these samples. I suspect that although it is not radiation damage occurring per se, the radiation is driving some of the kinetics in these samples. Such an effect has been seen in: Beam-induced atomic motion in alkali borate glasses, Holzweber et al, Phys. Rev. B 100, 214305 (2019). The transmission data shown in the rebuttal are also not what one would expect, I would expect them to follow Poisson statistics.

Having said this, I think the explanation in terms of curved stacked layers is a reasonable suggestion to explain parts of the data. As such it deserves to be published. The explanation may be incorrect, but it is not obviously so and this paper should lead (perhaps incite is a better word) to further work on the systems studied here.

Response to Reviewers for manuscript:

Ion complexation waves emerge at the curved interfaces of layered minerals

Authors: Michael L. Whittaker, David Ren, Colin Ophus, Yugang Zhang, Laura Waller, Benjamin Gilbert, Jillian F. Banfield

Reviewer #1

I appreciate the authors efforts to improve the manuscript. But I still do not understand the structure seen in the XPCS time-time correlation plots. For dynamical samples at steady state conditions (e.g., diffusion of spheres in a liquid) the time-time correlation patterns do not change with time. Relaxation dynamics of a glass should lead to slow variations on the time-time correlation functions.

- This is precisely what we show in Figure 1a for the case of Na-Mt in the absence of background electrolyte, which ‘lead[s] to slow variations on the time-time correlation functions’ over time.
- As we state in line 71:
 - Na-Mt in ultrapure water exhibited structural correlations that increased steadily over time ($\chi \rightarrow 1.2$), shown in Fig. 1a for the case at $q = 0.0025 \text{ \AA}^{-1}$ (250 nm, full range of q in Fig. S3).
- We provide what we believe to be the most accurate mechanistic description of this process for layered minerals to date in line 81
 - We find that layers have effectively zero mobility at distances below 58 nm, as determined from relaxation times extracted from the one-time $g_2(q, t)$ autocorrelation function derived from $\chi(q, t_1, t_2)$ (Fig. 1g, h, Supplementary Information). Thus, the gelling behavior in Fig. 1a corresponds to the gradual alignment of the nematic director field (Fig. 1c) for layers with translational diffusivities, D , of $1.7 \times 10^3 \text{ \AA}^2\text{s}^{-1}$ (Fig 1h, Fig. S3) above 58 nm separation and essentially zero at smaller interlayer distances where EDL repulsion restricts layer motion.
- Therefore, what the reviewer expects to see is in fact described in great detail at the outset of the manuscript.
- Because the reviewer does not acknowledge the importance (or even existence!) of Fig. 1a, they ascribe ‘dramatic behavior’ observed in the presence of background electrolyte to X-ray beam damage. But if the reality of Fig. 1a were appreciated, ‘dramatic behavior’ in the presence of background electrolyte can be very easily contextualized within the model we present (and cannot be explained by any model of beam damage that we are aware of).

The only cases for which I have seen such dramatic behavior in time-time correlation functions is for samples that are driven out of equilibrium by an event (i.e., quenching the sample temperature to drive a phase transition) and then the evolving time-time correlation data after the event reflects the subsequent evolution of the sample. In this experiment, the only event imposed on the sample (to my understanding) is the illumination of the X-ray beam.

- We believe the reason that the Reviewer’s assumptions are challenged here is because the energy scales that cause clay behavior to be driven from equilibrium are much smaller than many other systems, and therefore that the timescales on which structural evolution occurs are much longer. Picking up the sample capillary and inverting it to load it in the sample holder are sufficient to drive the sample out of equilibrium. It may then take days/weeks/months to re-equilibrate

- We cite evidence for the time and energy scales relevant to clay systems in our manuscript and explicitly in response to this same point in the Rebuttal. Why are the ample data that exist in the literature on this point and raised specifically by us not addressed at all by the Reviewer?

The data shown in the rebuttal appear to be consistent with this picture. The plots of total transmitted intensity vs. time in [Rebuttal] Figures 3-5 show that this signal systematically increases with time.

- We report only the 4% transmission data from Rebuttal Fig. 3 in our manuscript, which does not show a systematic increase in intensity with time, so the objections by the Reviewer on this point don't have anything to do with data presented in the manuscript.
- We show cases in which we damage the sample only to support our rationale for reporting data which do not show features of damage.
- We show that beam damage *can* indeed occur in these systems. Nonetheless, as Fig. S3 in our Supporting Information very clearly demonstrates, the dynamics calculated from these 'damaged' samples could not possibly reproduce the behavior observed in samples with different electrolyte concentrations or different cations if we irradiated the samples for one million years.
- The quantitative nature of our predictions cannot be emphasized enough. It is trivially easy to invoke damage. It is another thing to quantify the observed behavior and test it against a physical model. We would have been happy to test our data against a competing quantitative model for damage, or any other model for that matter, had one been suggested.
- After many rounds of requesting a quantitative model, test, or even a citation, including very explicitly in our Rebuttal, we have not been provided with one.
- The following question must be answered by the Reviewer if their claims are to carry any weight: through what damage mechanism can the observed dynamics depend on the (equilibrium) hydration energy of the cation (Fig. S3) over 6 orders of magnitude, while the observed effects of irradiation change the dynamics by < 1 order of magnitude?

The data also show that the fractional change in intensity becomes larger both for increasing beam flux (from 4% to 100%) and for increasing exposure times (e.g., 1 ms to 100 ms). This suggests that there are significant changes in the sample structure that are driven by the X-ray beam.

- The X-ray dose increases over time. Any process that also changes over time necessarily changes in proportion to increasing X-ray dose, but that does not establish a causal relationship that uniquely points to beam damage as the sole driver of the process.
- The transmission data we present in Rebuttal Fig. 3-5 are not consistent with any model of damage accumulation that we are aware of. Which model of damage is the Reviewer relying on to assert that these trends are indicative of damage? If 'damage' can be anything that is unfamiliar, then there is no recourse to combat claims of damage whenever something new is discovered.
- After the fourth round of review, saying that a very unfamiliar trend 'suggests' unequivocally that something is being driven by the X-ray beam is just woefully insufficient. The changes in intensity are not linear but the accumulated X-ray dose is, so what functional dependence on X-ray dose is expected from 'damage'? We provide a model that explains non-linear change in the sample structure over time. Which model of damage explains the data better?

- If the Reviewer were to temporarily accept our premise and explore the consequences, they would find that there are explanations of the observed trends in intensity that are also completely consistent with the data we present. Our explanations also benefit from being supported by quantitative models and unprecedentedly high quality imaging studies as corroborating evidence. What if variations in intensity are due to the fact that clay layers are moving dynamically within the solution, and doing so in a correlated way, that may also cause them to redistribute outside the beam path for a period of time?

I believe that this is the source of the complex changes seen in the time-time correlation functions, as opposed to any intrinsic dynamics associated with the electrostatic adsorption of ions.

- Ultimately, we see that this is a matter of belief.
- As we've gone to great lengths to convey to the Reviewer (and which is not a novel or controversial aspect of our work), there are many forces besides 'electrostatic' that control ion adsorption. If one wishes to attempt to rationalize our observations with 'electrostatic adsorption of ions', it is no surprise that they will be left with questions.
 - From line 61:
 - "We find that layer dynamics in montmorillonite (Mt) suspensions at low-electrolyte conditions are largely consistent with [electrostatic] DLVO theory, while qualitatively new behavior that emerges at elevated electrolyte concentrations."
 - Returning to our response to the first point raised by the Reviewer in this document, when one ignores the presence of data that is adequately explained by well-established theory *and* the Reviewer's own expectations (Fig. 1a), then it is no surprise that novel observations that are shown to violate these assumptions in a very specific and quantifiable way don't make sense.
- We would implore the Reviewer to suggest a specific experiment that could be conducted to test their hypothesis. We have heard over 4 rounds of review that the Reviewer thinks that the dynamics we observe are beam damage. At each stage we have attempted to address the Reviewer's concern with more data, including data that they have asked for explicitly. However, this clearly wasn't adequate. It is hard to avoid the conclusion that after seeing the manuscript for a fourth time, the avoidance of acknowledging Fig. 1a, for example, (as discussed above) is something other than seeing only that which serves a specific purpose.
- What test that we have not already conducted would be sufficient to convince the Reviewer? We asked this question in the Rebuttal and do not see a suggestion of one here.

I do not want to be unfair to the authors. There is a disagreement with respect to the significance of the results. I suggest that another reviewer look at the results to provide another opinion.

Reviewer #2

This is not an easy paper to read.

- Regrettably, our many attempts to address concerns while adhering to spatial requirements have negatively impacted the narrative flow of the manuscript. We have made tracked changes to the text to improve readability without making any substantive changes to the concepts or descriptions.

The data clearly show that there is an electrolyte dependent change in the samples. This is shown clearly by the qualitative difference in the XPCS data and the detailed analysis of the cryoEM results. It is also clear that there is a non-equilibrium aging behavior of the low concentration data as seen by the increasing correlation times. One question is "what starts this behavior?". Is it something in the preparation of the samples or is it their introduction into the x-ray beam. This increasing correlation time with aging time does not seem so evident in the higher concentration data, at least when I try to picture "averaging" over the fast structures to see the long time behavior. Clearly, something unusual is happening in these samples. I suspect that although it is not radiation damage occurring per se, the radiation is driving some of the kinetics in these samples. Such an effect has been seen in: Beam-induced atomic motion in alkali borate glasses, Holzweber et al, Phys. Rev. B 100, 214305 (2019).

- This is an interesting reference on a different system (the two alkali cations (Rb, Cs) we do not investigate in our study (Li, Na, K), and in a borosilicate glass and not in aqueous solution).
- From this reference:
 - “In these systems, there is a clear dependence of the atomic motion on the incident x-ray flux, which implies that the observed dynamics is driven by the absorbed x-ray intensity. Still, no significant structural modification of the sample is evident. This is in stark contrast to studies on crystalline alloys or metallic glasses, where such an effect is not seen”
- Our data do not show a strong flux dependence (Fig. S3, for which data with different total dose were also acquired with different dose rates), and therefore, likely fall into the category of systems ‘where such an effect is not seen’.
- In fact, the relative fluxes we examine vary from 0.4% to 100%, while the reference study varies only from 23% to 100%. The actual fluxes in our system are also approximately 20x lower ($\sim 10^{11}$ photons/s in both cases, but our beam size is $100 \mu\text{m}^2$ while the reference uses a beam size of $5 \mu\text{m}^2$). Thus, we examine a much larger range of incident flux *that are all well below the range of fluxes observed to cause beam induced effects* and find very little effect.
- Furthermore, the beam energy used in the referenced study was 13 keV compared to 9.65 keV in our study. Our beam energy exceeds the absorption edge of all the elements in our systems (K being the heaviest, with a K-edge at 3.6 keV), so absorption is entirely possible but occurs with very low probability, given the roughly exponential dependence of absorption on the difference between the edge and beam energies. The beam energy of 13 keV used in the referenced study was much closer to the Rb K edge (15.2 keV), which showed the strongest ‘damage’ effects. Thus, the reported phenomenon could be a resonant effect related to the proximity of absorption edge energies to the incident beam energy. No resonant effects are expected in our system.

- Beam-driven atomic dynamics observed in Rb/Cs borate glasses are quantified for scattering vectors, q , of approximately $1.5\text{-}2.0 \text{ \AA}^{-1}$. This is three orders of magnitude larger (corresponding to structures three orders of magnitude smaller) than the scattering vectors that we examine in our work (which looks at the motion of clay layers). Therefore, while we cannot completely rule out the possibility that the beam interacts differently with different alkali cations in our system (since we do not directly probe those length scales), we can confidently conclude that our general model still holds because *the motion of ions at the interface is driving dynamics in clay layers many orders of magnitude larger*. Testing to what extent this process is exacerbated by the X-ray beam may be an interesting follow on study (although we stress again that Fig. S3 and Fig. S7 indicate that the flux dependence is quite small, if present at all, as discussed above), but does not alter the conclusions we present.
- Nonetheless, this is exactly the type of quantitative model that we wished to have been able to test against in response to previous objections raised by other reviewers. From the above discussion it is very clear what ‘damage’ looks like, and how to test for it, using quantitative descriptions of the relevant process variables such as beam flux and energy.

The transmission data shown in the rebuttal are also not what one would expect, I would expect them to follow Poisson statistics.

- While we do not have a quantitative model for the statistics of the transmission data, we believe that our observations are consistent with a dynamic system of clay layers whose collective behavior is not stochastic (and therefore may not follow Poisson statistics). For example, if variations in intensity are due to the fact that clay layers are moving dynamically within the solution, and doing so in a correlated way, they may redistribute outside the beam path for a period of time that is on the same timescale as the experiment, altering the transmission in a way that is unrelated to beam-induced effects.

Having said this, I think the explanation in terms of curved stacked layers is a reasonable suggestion to explain parts of the data. As such it deserves to be published. The explanation may be incorrect, but it is not obviously so and this paper should lead (perhaps incite is a better word) to further work on the systems studied here.

- We don’t wish to incite, but we do believe that our results are provocative and hope that they catalyze future work.